# MRLIHT: Mobile RFID-Based Localization for Indoor Human Tracking

**DOI:** 10.3390/s20061711

**Published:** 2020-03-19

**Authors:** Qian Ma, Xia Li, Guanyu Li, Bo Ning, Mei Bai, Xite Wang

**Affiliations:** College of Information Science and Technology, Dalian Maritime University, Dalian 116026, China; lixia_email@163.com (X.L.); liguanyu@dlmu.edu.cn (G.L.); ningbo@dlmu.edu.cn (B.N.); baimei@dlmu.edu.cn (M.B.); wangxite@dlmu.edu.cn (X.W.)

**Keywords:** RFID, mobile reader, fixed tag, indoor localization, human tracking

## Abstract

Radio Frequency Identification (RFID) technology has been widely used in indoor location tracking, especially serving human beings, due to its advantage of low cost, non-contact communication, resistance to hostile environments and so forth. Over the years, many indoor location tracking methods have been proposed. However, tracking mobile RFID readers in real-time has been a daunting task, especially for achieving high localization accuracy. In this paper, we propose a new Mobile RFID (M-RFID)-based Localization approach for Indoor Human Tracking, named MRLIHT. Based on the M-RFID model where RFID readers are equipped on the moving objects (human beings) and RFID tags are fixed deployed in the monitoring area, MRLIHT implements the real-time indoor location tracking effectively and economically. First, based on the readings of multiple tags detected by an RFID reader simultaneously, MRLIHT generates the response regions of tags to the reader. Next, MRLIHT determines the potential location region of the reader where two algorithms are devised. Finally, MRLIHT estimates the location of the reader by dividing the potential location region of the reader into finer-grained grids. The experimental results demonstrate that the proposed MRLIHT performs well in both accuracy and scalability.

## 1. Introduction

Radio Frequency Identification (RFID) [1,2,3] is a technique of automatic identification using radio frequency transmission. RFID technology is widely used in many real-world applications, e.g., real-time tracking, activity recognition, etc, and impacts almost all aspects of people’s daily life. An RFID system consists of readers and tags, where readers communicate with tags through radio signals. RFID tags are categorized as either active or passive. Active tags are powered by on-board batteries and broadcast their signals continuously, while passive tags are powered by the electromagnetic energy transmitted from RFID readers. Since the maturely developed GPS-based localization systems do not perform well for indoor location tracking applications, RFID-based localization technologies have become popular in several indoor tracking schemes. In this paper, we aim to propose an effective RFID-based localization approach for indoor human tracking, which can be pervasively used in the large-scale and high-accurate tracking applications.

Over the years, many RFID-based indoor human/object tracking methods have been proposed. They can be categorized as fixed RFID (F-RFID) model based and mobile RFID (M-RFID) model based tracking, whereas in the F-RFID (i.e., fixed RFID) model, RFID readers are pre-deployed at some fixed points in the monitoring area and RFID tags are attached on the moving objects, while in M-RFID (i.e., mobile RFID) model, RFID readers are equipped on the moving objects and RFID tags are fixed deployed at some points in the monitoring area. Generally, the two models are used for different application scenarios. In large-scale monitoring applications, since the cost of RFID readers are very high, the deployment of fixed readers are generally sparse, which results that the location information of moving individuals (mobile tags) are discontinuous. In other words, there are phenomena of location information missing. To tackle this issue, some existing works estimate the missing location information based on the pre-defined motion model. Nevertheless, in most cases, the motion of individuals is arbitrary and cannot be predicted. On the other hand, to alleviate the sparseness of RFID readers, some existing works use the reference tags to instead replacing more readers. However, the localization accuracy is unsatisfactory (incurring meters of localization errors). In addition, to estimate the locations of moving human/objects based on the F-RFID model, the positions of fixed readers which are stored in the external storage medium (e.g., databases or files) are needed. The location information requirements incur extra query costs, and thereby reduces the tracking efficiency.

Therefore, in this paper, we propose a new Mobile RFID (M-RFID) based Localization approach for Indoor Human Tracking (MRLIHT). By deploying a large number of fixed and passive RFID tags in the monitoring area, the locations of moving individuals equipped with RFID readers are determined in real-time. The advantages that we adopt the M-RFID model and passive tags are as follows.

Lower costs. Compared with active tags, the light-weight and small-volume passive tags without on-board batteries have longer lifetime and lower costs. Moreover, there is almost no maintenance cost [4]. On the other hand, in the scenario of human indoor location tracking, the RFID readers are held by human beings and can be reused many times, which are economical in a long-term perspective.Smaller localization errors. First, the sensing range of passive tags is smaller than that of active tags. During object localization, the candidate locations of an individual filtered by passive tags are more fine-grained, and thus the localization accuracy is improved. Second, by deploying passive tags intensively to cover the whole monitoring area, the location of a moving individual can be estimated continuously based on the location information of multiple tags received by the reader equipped on the moving object. Then the problem of location information missing introduced earlier can be effectively solved.Avoiding extra queries. With M-RFID model, RFID tags are fixed deployed in the monitoring area, rather than attaching on the moving individuals. Hence, the location information of passive tags can be written in their own memories and read by RFID readers directly. Therefore, the extra queries for location information are avoided.

For an individual, MRLIHT estimates its location by generating the response regions of tags detected by the reader attached to the individual first. Secondly, potential location region of the individual (reader) is determined based on those response regions. Finally, the final location of the individual is estimated with high accuracy and efficiency. The major contributions made in this paper are summarized as follows.

We point out the pitfalls of existing works for large-scale and high-accurate tracking tasks. Accordingly, we develop an M-RFID based localization approach for indoor human tracking, named MRLIHT, to support high-quality moving humans tracking.By considering the influence of environmental noises, we extend the response region of an RFID tag to an RFID reader from a circle to a circular, which improves the localization accuracy.In MRLIHT, there is no pre-defined motion model for moving individuals, which are more suitable for real-world applications.For the fixed passive tags, the location information is written in their own memories directly. The RFID readers can obtain the location information of surrounding tags without extra query, which improves the location estimation efficiency.Finally, extensive experiments are conducted to evaluate the performance of the proposed approach.

The organization of the remaining paper is as follows. Section 2 discusses the related work. Section 3 presents the problem definition. Section 4 introduces the proposed MRLIHT system and Section 5 shows the experimental results. Finally, Section 6 concludes the paper.

## 2. Related Work

With the rapid development of big data [5], the idea of indoor location tracking based on RFID technology is widely used in many real applications. The traditional RFID-based indoor location tracking approaches can be classified into two classes: fixed RFID (F-RFID) based tracking and mobile RFID (M-RFID) based tracking.

Most of the existing works [6,7,8] are proposed for F-RFID model. In work [6], a localization framework, named LADNMARC, is proposed, where an indoor moving object is located based on the signal strengths received by nearby RFID readers. To improve the positioning accuracy while saving cost, LADNMARC proposes reference tags to instead replacing more readers. However, the deployment of RFID readers and reference tags are very strict in LADNMARC. RADAR [7] builds a signal propagation model by considering the impact of walls in indoor environments. It determines the location of an indoor object by combining the signal propagation model with empirical measurements. An indoor tracking method for navigating visually impaired people is proposed in [8]. The solution takes the signal strengths received by RFID readers as an observation vector, and determines the location of a user based on the Bayesian Decision Theory. The methods introduced above are active-tag-based systems, where the basic idea of location tracking is to make use of the signal strengths received by RFID readers and broadcast by active tags. However, since the sensing range of active tags is up to hundreds of meters, the localization error is large.

In addition, some works make use of passive tags to locate indoor objects based on the F-RFID model. PinIt [9] is a fine-grained RFID localization scheme proposed for nonline-of-sight scenarios. BackPos [10] is another fine-grained backscatter positioning technique based on the detected phase model. To reduce the labor and time cost of location fingerprinting, a cost-effective neighborhood graph-based localization system is proposed in [11]. However, all approaches mentioned above are mainly used for locating the static RFID tags and do not perform well in tracking the moving objects. In [12], an RFID data cleaning approach that supports location query is proposed. It estimates the location of moving objects based on Bayesian inference and taking advantage of duplicate readings. Since the Received Signal Strength Indicator (RSSI), which is widely adopted in active-tag-based tracking methods, is not available for passive tags, PassTrack [4] proposes a reader detection model based on the reading rates of passive tags. Then the location of a moving object is estimated based on the reader detection model. The work [13] studies the problem of cleaning RFID data streams for object tracking. To tracking objects in mobile environments, a probabilistic inference model is proposed. However, all the passive-tag-based indoor tracking algorithms introduced above need to be assigned a motion model to moving objects. In practice, the movement of indoor objects are generally arbitrary and unpredictable. OTrack [14] is proposed to track the luggage attached with tags in the airport based on the temporal correlation among the communications between the RFID readers and tags. However, such a scheme usually requires specific application scenarios. Tagoram [15] achieves the goal of tracking mobile RFID tags in real-time by leveraging the phase value of the backscattered signal, which suffering the problem of multipath reflections. MobiTagbot [16] constructs a holography to locate RFID tags using antenna movements and frequency hopping. However, MobiTagbot is time-consuming for large-scale monitoring areas and impractical for real-time localization.

By comparison, the M-RFID based indoor tracking method is few. In [17], the authors propose to use relatively fixed tags to locate a moving human/object with a mobile reader. Considering the location uncertainty of the moving objects, the work proposes a probabilistic model to locate the moving objects. However, the algorithm is also based on the assumption that the motion model of moving objects is known, which limits its applications. In work [18], the proposed data cleaning approach for mobile RFID data streams is capable of large tracking and monitoring environments. The rationale of the proposed approach is similar to those methods based on the F-RFID model and only considers the case of a single reader. The works [19,20] use the memory of passive tags to encode and store the information of moving objects. Then the proposed algorithms estimate the locations of moving objects through creating an information-gradience. However, the approaches are time-consuming and not capable of real-time localization. In work [21], a phase-based localization algorithm is proposed to locate a device moving equipped with a UHF RFID reader.UHF (Ultra High Frequency) RFID reader is a kind of RFID reader where the frequency range is from 300 MHz to 1 GHzIn this work, the mobile device is supposed to move along a known path (e.g., x-coordinate) and its location at an assigned temporal can be estimated by using the phase differences between the RFID reader and reference tags besides the path. Moreover, the moving speed of the mobile device and its location on other coordinates (e.g., y-coordinate and z-coordinate) are also needed, which is difficult to obtain in practice. Some works, e.g., STPP [22] and TagSort [23] propose the concept of relative localization. They determine the order (or relative location) of fixed tags by exploring the change of phase information of tags when a moving reader approaches or moves away. However, the proposed methods focus on obtaining the order of tags rather than the absolute location of tags. Moreover, they are proposed for positioning the static objects which are not competent for the moving objects. On the other hand, there are some works are suitable for both F-RFID and M-RFID, e.g., *k*-nearest-neighbor-based localization [24,25] and multilateration-based localization [26], but their localization accuracy are needed to be improved. In this paper, we propose a new approach based on the M-RFID model to tracking indoor humans with high-accuracy and in real-time.

## 3. Problem Statement

Suppose there are *n* RFID tags $T=\left\{T_{1},T_{2},\cdots ,T_{n}\right\}$ and *m* RFID readers $R=\left\{R_{1},R_{2},\cdots ,R_{m}\right\}$ in the monitoring area, where the RFID tags are deployed at some fixed points and RFID readers are handheld or equipped on the moving individuals. The main goal of this work is to determine the locations of moving individuals in real-time. In the following, we first formally define some terms used throughout the paper.

**Definition** **1.**
*A reading, denoted by $D=\left(Rid,Tid,Tloc,Time,Count\right)$, is a data tuple with information of an RFID tag by an RFID reader, where Rid and Tid are the unique identifications of the reader and tag respectively, Tloc is the location of the tag, Time is a unit time point and Count is the number of times the tag detected by the reader in unit time.*


In this paper, we use 2D coordinate to denote the location of an RFID tag (or an RFID reader), i.e., the location of an RFID tag $T_{j}$ (or an RFID reader $R_{i}$) can be denoted by $T_{j}.loc=\left(T_{j}.X,T_{j}.Y\right)$ (or $R_{i}.loc=\left(R_{i}.X,R_{i}.Y\right)$). Moreover, it is notable that, in the F-RFID model, the reading of a tag by a reader is usually denoted by a triple $\left(Rid,Tid,Time\right)$ without the information about tag location and the number of detection times. It is because, with the F-RFID model, RFID tags dynamically move with the moving objects (human beings), while the memories of RFID tags are limited and unavailable for being written frequently. By comparison, in M-RFID model, the RFID tags are fixed and thus their locations can be written into their own memories once only. In addition, since RFID readings are of low quality, the read frequency of RFID readers is high. Hence, an RFID tag can be detected multiple times by an RFID reader in unit time. To save storage space and reduce the data complexity, we aggregate multiple readings of a tag by an RFID reader in unit time into a single reading by adding the Count attribute (as introduced in Definition 1).

**Example** **1.**
*Figure 1 is the plan of a large-scale indoor exhibition area where many spots can be visited. To tracking visitors accurately, a large number of RFID passive tags are intensively deployed in the monitoring area (denoted by the red circle in the figure) and RFID readers are equipped on the moving objects (i.e., the visitors). Note that with the development of hardware, the portable and small RFID reader has been widely used in many real applications. Specifically, the distribution of tags detected by reader $R_{3}$ at time $t_{1}$ is shown in the right subfigure where ten tags, i.e., $T_{1}∼T_{10}$, are detected by $R_{3}$ simultaneously. Suppose the reading of a tag (e.g., $T_{4}$) by reader $R_{3}$ at time $t_{1}$ is $D_{3,4}=\left(R_{3},T_{4},\left(2,8\right),t_{1},7\right)$. It means that at time $t_{1}$, tag $T_{4}$ is detected seven times by reader $R_{3}$.*


As introduced earlier, the passive RFID tags are powered by the energy from RFID readers instead of on-board batteries. An RFID reader emits an electromagnetic energy field of a few feet, and any RFID tag in the vicinity may receive the energy and response its information.

**Definition** **2.**
*With the M-RFID model, when a fixed tag responds to a moving reader, the region where the reader may locate in is the Response Region (RR) of the tag to the reader.*


Moreover, a reader may detect multiple tags simultaneously due to the intensive deployment of tags and large sensing range of the reader.

**Definition** **3.**
*The Potential Location Region (PLR) of a reader is the overlap region of tag response regions which are generated by the tags detected by the reader simultaneously.*


On the other hand, even though an RFID tag locates in the sensing range of an RFID reader, it may not respond to the reader successfully, as the energy gathered by the tag may not enough to respond. In other words, an RFID tag is detected by an RFID reader with a certain probability, rather than definitive.

**Definition** **4.**
*The reading rate of an RFID tag by an RFID reader, denoted by τ, is the probability of the tag is detected by the reader.*


Since the Received Signal Strengths Indicator (RSSI) measurement which usually adopted in active-tag-based systems, is not available for passive tags, we aim to leverage the reading rate to obtain the location information of moving objects (persons). To achieve this goal, we propose an M-RFID based Indoor Location Tracking (MRLIHT) system. At time *t*, given the readings of tags by each reader, MRLIHT aims to tackle the following issues:Generating the response region of each tag detected by a reader simultaneously.Determining the potential location region (PLR) for each RFID reader.Estimating the location of each RFID reader (i.e., moving object) precisely.

Finally, the locations of moving persons at each time point are returned with high-accuracy and in real-time.

## 4. MRLIHT System

In this section, we introduce the proposed MRLIHT which consists of three components: RR generation, PLR determination and location estimation to tackle the issues mentioned above.

### 4.1. RR Generation

As introduced earlier, we use the reading rate of a passive tag by a reader to determine the location of an indoor moving individual. Specifically, there are two ways to estimate the reading rate.

One way is to use the response count in a fixed number of interrogation cycles sent from the reader. In this work, we estimate the reading rate of a tag $T_{i}$ by a reader $R_{i}$ as
(1)$$\tau_{ij}=\frac{{Count}_{ij}}{C_{i}},$$
where ${Count}_{ij}$ is the number of times tag $T_{j}$ detected by reader $R_{i}$ and $C_{i}$ is the total number of interrogation times of reader $R_{i}$ in unit time. For example, if a reader receives responses from a tag in 30 out of 100 interrogation times in unit time (e.g., 1 second), then the reading rate of the tag by the reader is 0.3.

Another way is to estimate the reading rate of a tag by a reader based on the reader detection model, where the reading rate is estimated by a function about the distance between the tag and the reader. In this work, we adopt the Sigmoid model to quantify the reading rate with distance proposed in [4], because the reader detection model is more accurate than the traditional 3-state model [12]. For an RFID tag $T_{j}$, the reading rate of $T_{j}$ by an RFID reader $R_{i}$ is shown below
(2)$$\tau_{ij}\left(l_{ij}\right)=\frac{1}{1+e^{a_{i}l_{ij}+b_{i}}},\;\left(a_{i}>0,b_{i}>0\right).$$

In Equation (Equation 2), $a_{i}$ and $b_{i}$ are the parameters of reader $R_{i}$ and can be learned from the readings of the detected tags by the reader $R_{i}$. The variate $l_{ij}$ is the distance between the reader $R_{i}$ and tag $T_{j}$. Since the reading rate of tag $T_{j}$ by reader $R_{i}$ can be obtained through the readings of $T_{j}$ by $R_{i}$ based on Equation (Equation 1), the distance $l_{ij}$ between $R_{i}$ and $T_{j}$ in Equation (Equation 2) can be computed below
(3)$$l_{ij}=\frac{1}{a_{i}}\left[ln\left(\frac{1}{\tau_{ij}}-1\right)-b_{i}\right].$$

As introduced earlier, the locations of readers and tags are denoted by the points with 2D coordinate. The Euclidean distance between $R_{i}$ and $T_{j}$ is $l_{ij}=\sqrt{{\left(R_{i}.X-T_{j}.X\right)}^{2}+{\left(R_{i}.Y-T_{j}.Y\right)}^{2}}$. Then we have
(4)$${\left(R_{i}.X-T_{j}.X\right)}^{2}+{\left(R_{i}.Y-T_{j}.Y\right)}^{2}=l_{ij}^{2},$$
where $l_{ij}$ can be computed based on Equation (Equation 3), the location of tag $T_{j}$, i.e., $T_{j}.loc=\left(T_{j}.X,T_{j}.Y\right)$ is known and the location of reader $R_{i}$, i.e., $R_{i}.loc=\left(R_{i}.X,R_{i}.Y\right)$ is the goal we aim to determine. From Equation (Equation 4), we can know that the region where reader $R_{i}$ may locate in, i.e., the response region (RR) of tag $T_{j}$ to reader $R_{i}$, is the circle with center $\left(T_{j}.X,T_{j}.Y\right)$ and radius $l_{ij}$.

In addition, due to the influence of environmental noises and the antenna direction settings of the reader, the reading rate of a tag by a reader has certain errors. Based on the theory of radio propagation, for the same source (i.e., reader), the error of signal strength (denoted by ϵ) received by various receivers (i.e., tags) follow the same statistical law, i.e., the Gaussian error model $\varepsilon ∼N\left(\mu ,\sigma^{2}\right)$ [27]. Therefore, in this paper, we suppose that the error of $R_{i}$’s reading rate follows the Gaussian distribution with mean 0 and variance $\sigma_{i}^{2}$, i.e., $\varepsilon_{i}∼N\left(0,\sigma_{i}^{2}\right)$, where the mean is 0 as we believe that the methods do not make errors on purpose and the variance $\sigma_{i}^{2}$ can be estimated based on the readings of the tags by the reader. Therefore, given a confidence value 1−α, we have
$$p\left(-N\left(1-\alpha /2\right)<\left|\frac{\varepsilon_{i}}{\sigma_{i}}\right|<N\left(1-\alpha /2\right)\right)=1-\alpha .$$

Adding the error $\varepsilon_{i}$ on the reading rate $\tau_{ij}$, the range of reading rate of tag $T_{j}$ by reader $R_{i}$ is
(5)$$\left(\tau_{ij}-\sigma_{i}N\left(1-\alpha /2\right),\tau_{ij}+\sigma_{i}N\left(1-\alpha /2\right)\right).$$

Combining Equation (Equation 5) with Equation (Equation 3), the range of distance between reader $R_{i}$ and tag $T_{j}$ is $l_{i}\in \left[l_{min},l_{max}\right]$, where
$$l_{min}=\frac{1}{a_{i}}\left[ln\left(\frac{1}{\tau_{ij}+\sigma_{i}N\left(1-\alpha /2\right)}-1\right)-b_{i}\right],$$
$$l_{max}=\frac{1}{a_{i}}\left[ln\left(\frac{1}{\tau_{ij}-\sigma_{i}N\left(1-\alpha /2\right)}-1\right)-b_{i}\right].$$

Finally, the response region (RR) of tag $T_{j}$ to reader $R_{i}$ becomes a circular region with center $\left(T_{j}.X,T_{j}.Y\right)$ and radius $l_{min}$ and $l_{max}$.

**Example** **2.**
*For the reader $R_{3}$ and tag $T_{4}$ mentioned in Example 1, suppose the reading rate of $T_{4}$ by $R_{3}$ is $\tau_{3,4}=0.8$, the variance $\sigma_{3}$ of error $\epsilon_{3}$ is $\sigma_{3}=0.1$ and the confidence value is 1−α=0.9. Moreover, let the parameters of reader detection model are $a_{3}=9,843$ and $b_{3}=-16.2015$. We can compute that $l_{min}=1.379$ and $l_{max}=1.584$. Then the response region of tag $T_{4}$ to reader $R_{3}$ is shown in Figure 2a.*


### 4.2. PLR Generation

Thus far, given an RFID reader $R_{i}$ and an RFID tag $T_{j}$ detected by $R_{i}$, we can obtain the RR (response region) of $T_{j}$ to $R_{i}$. Next, we aim to determine the potential location region (PLR) of a reader by employing the RRs of tags detected by the reader simultaneously.

#### 4.2.1. Exact Solution

Based on Definition 3, the PLR (potential location region) of $R_{i}$ is derived by taking the overlap of RRs of multiple tags detected by $R_{i}$ simultaneously.

**Example** **3.**
*(Example 2 Continued) The PLR of reader $R_{3}$ derived by the RRs of three tags it detected simultaneously is shown in Figure 2b (the polygon surrounded by the blue curve). Since it is difficult to show the intersection of RRs of ten tags detected by reader $R_{3}$ simultaneously, we only show the PLR of $R_{3}$ derived by RRs of three tags it detected simultaneously. Moreover, Note that in practice, a reader may detect a large number of tags simultaneously due to the intensive deployment of passive tags.*


Algorithm 1 shows the pseudo-code of the exact solution of PLR (potential location region) generation. As shown, for each reader $R_{i}$, we need to compute the RR (response region) of each tag $T_{j}$ detected by $R_{i}$ simultaneously, and taking the overlap of multiple RRs to generate the PLR of $R_{i}$. Since a tag may also be detected by multiple readers, the time complexity of the exact PLR generation solution is $O\left(mn\right)$ where *m* and *n* are the numbers of readers and tags in the monitoring area, respectively.
**Algorithm 1:** PLRGenExact(T, R, D, $a_{i}$, $b_{i}$, α).
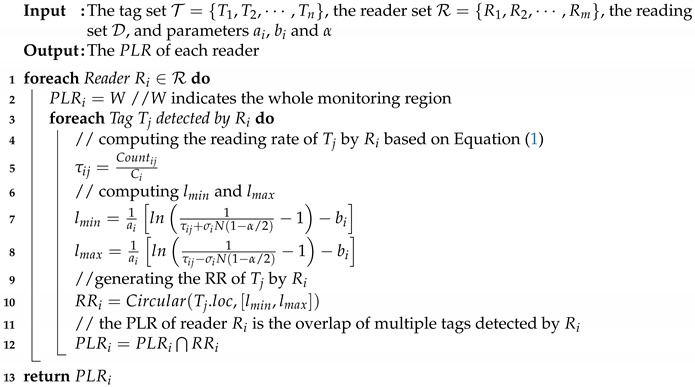


#### 4.2.2. Approximate Solution

The exact solution may be inefficient for large-scale tracking applications where there are hundreds of thousands or millions of passive tags deployed in the monitoring area. Under this scenario, a large number (dozens or even hundreds) of tags may be detected by a reader simultaneously. As a result, the time complexity of PLR generation is extremely high by taking the overlap of multiple detected tags. Moreover, due to the intensive deployment, some RFID tags are close to each other, which results that the RRs of these tags are almost the same. To avoid redundant computation, we further propose an approximate solution to generate the PLR for each reader. The approximate solution consists of two steps: (1) tag group generation and (2) PLR determination. Next, we introduce each step in detail.

First, we divide the RFID tags into tag groups defined below.

**Definition** **5.**
*Tag Group. A tag group G contains a collection of tags satisfying that*
*(i)* 
*$\forall T_{j}\in G$, $\exists T_{k}\in G$, $d\left(T_{j},T_{k}\right)\leq \delta_{d}$ where $d\left(T_{j},T_{k}\right)$ is the distance between $T_{j}$ and $T_{k}$ and $\delta_{d}$ is the distance threshold.*
*(ii)* 
*$\left|G\right|\geq \delta_{s}$ where $\left|G\right|$ is the size of the tag group, i.e., the number of tags contained in G, and $\delta_{s}$ is the size threshold.*



Note that we use the distance threshold $\delta_{d}$ to control the extension ability of a tag group, i.e., more tags may be included in a tag group with a greater $\delta_{d}$. On the other hand, we use the size threshold $\delta_{s}$ to avoid that sparse tags which have big distance are included into a tag group under the scenario of great $\delta_{d}$.

Algorithm 2 shows the pseudo-code of the tag group generation, where the specific steps are as follows:

(s1) Initializing a candidate tag group *G* and generating a seed list (denoted by S) which is the copy of tag set T (Line2-3).

(s2) Selecting a tag $T_{j}$ randomly as seed and removing it from the seed list S (Lines 5 and 6).

(s3) Computing the distance between each tag $T_{k}\in T$ and the seed $T_{j}$ (Line 9). Adding $T_{k}$ into the candidate tag group *G* and removing $T_{k}$ from the tag set T, if the distance is smaller than the given distance threshold $\delta_{d}$. Otherwise, removing $T_{k}$ from the seed list S (Line 10-14).

(s4) Repeating steps (s2) and (s3) until the seed list is empty, which means the distances between the remaining tags in the tag set and any tag in the candidate tag group *G* are greater than $\delta_{d}$.

(s5) Adding the candidate tag group into the set of tag groups G (Line 15).

(s6) Repeating steps (s1)–(s5) until the tag set is empty, i.e., all tags are classified into a certain tag group.

(s7) Determining each size of the candidate tag group. If the size is greater than the give size threshold $\delta_{s}$, it is a real tag group. Otherwise, remove the candidate tag group from S (Line 16–18). Finally, returning the set of tag group G.
**Algorithm 2:** TagGroupGen(T, $\delta_{d}$, $\delta_{s}$).
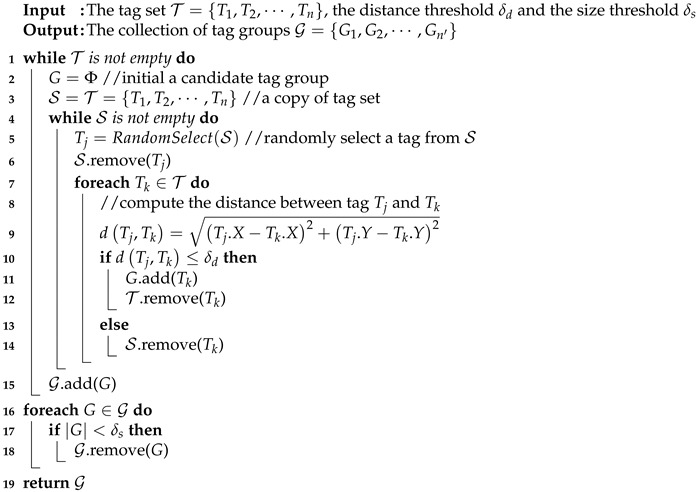


The time complexity of tag group generation is $O\left(n^{2}\right)$ and $O\left(n\right)$ in the worst case where each tag is considered as a tag group and in the best case where all RFID tags are considered as a tag group, respectively. Thus the average time complexity of tag group generation is $O\left(nlogn\right)$. Moreover, since our work is proposed based on the M-RFID model where RFID tags are fixed deployed in the monitoring area, the tag group generation only needs to conduct one time. On the other hand, it is notable that an RFID tag may not belong to any tag group. For such tags, they are considered independent and used to PLR determination with aggregated tags (to be introduced later) together.

After dividing tags into tag groups, we use the aggregated tags (generated based on the tag groups) to determine the PLR for each reader.

**Definition** **6.**
*Aggregated Tag. For a tag group G, the aggregated tag $T_{a}$ is the representation of G. Moreover, the reading of $T_{a}$ by a reader $R_{i}$ at a certain time point is computed based on the readings of tags by $R_{i}$.*


Given an RFID reader $R_{i}$, suppose the tag set detected by $R_{i}$ simultaneously (i.e., at a certain time point *t*), is $T_{i}=\left\{T_{1},T_{2},\cdots ,T_{q}\right\}$. We first determine each tag $T_{j}\in T_{i}$ belonging to which tag group. Then we generate the aggregate tags of corresponding tag groups used for $R_{i}$’s PLR generation. Since RFID tags are fixed, it is easy to locate tag $T_{j}\in T_{i}$ belongs to which tag group by building an inverted index. Specifically, during the tag group generation, we build an inverted index for each tag $T_{j}$ to record the tag belongs to which tag group. The structure of the inverted index is shown in Figure 3. Note that as mentioned earlier, an RFID tag may not belong to any tag group. Accordingly, in the inverted index, the tag points to ‘Null’ if it does not belongs to a tag group. Next, we introduce how to generate an aggregate tag for each tag group used for $R_{i}$’s PLR generation.

Suppose the tags detected by $R_{i}$ simultaneously belongs to *w* tag groups $G_{i}=\left\{G_{1},G_{2},\cdots G_{w}\right\}$ where $G_{k}=\left\{T_{k,1},T_{k,2},\cdots ,T_{k,q^{\prime }}\right\}$ ($T_{k,j}\in T_{i}$ and $q^{\prime }\leq q$) and the reading of $T_{k,j}$ by $R_{i}$ is $D_{k,j}=\left({Rid}_{k,j},{Tid}_{k,j},{Tloc}_{k,j},{Time}_{k,j},{Count}_{k,j}\right)$. We aim to obtain the reading of the aggregated tag corresponding to the tag group $G_{k}\in G_{i}$, denoted by $T_{a,k}=\left({Rid}_{a,k},{Tid}_{a,k},{Tloc}_{a,k},{Time}_{a,k},{Count}_{a,k}\right)$.

Since any tag $T_{i}\in T_{i}$ is detected by reader $R_{i}$ at a certain time point *t*, we have ${Rid}_{a,k}=R_{i}$, ${Time}_{a,k}=t$. In addition, we assign a unique ID for $T_{a,k}$ based on an independent number system. Next, we need to determine the location of the aggregated tag ${Tloc}_{a,k}$ and the number of detection times ${Count}_{a,k}$. In real applications, the distance between an RFID tag $T_{j}$ and a reader $R_{i}$ is smaller, the reading rate of $T_{j}$ by $R_{i}$ is higher, and thereby the location of the $T_{j}$ (used for determining the location of $R_{i}$) is more important. Therefore, we compute ${Tloc}_{a,k}$ and ${Count}_{a,k}$ by the weighted average of corresponding information of various tags, as shown below.
(6)$${Tloc}_{a,k}.X=\frac{\sum w_{k,j}\times T_{k,j}.X}{\sum w_{k,j}}$$
(7)$${Tloc}_{a,k}.Y=\frac{\sum w_{k,j}\times T_{k,j}.Y}{\sum w_{k,j}}$$
(8)$${Count}_{a,k}=\frac{\sum w_{k,j}\times {Count}_{k,j}}{\sum w_{k,j}}$$
(9)$$w_{k,j}=\frac{{Count}_{k,j}}{\sum {Cout}_{k,j}}$$

The pseudo-code of aggregated tag generation is shown in Algorithm 3.
**Algorithm 3:** AggregatedTagGen($R_{i}$, $T_{i}$)
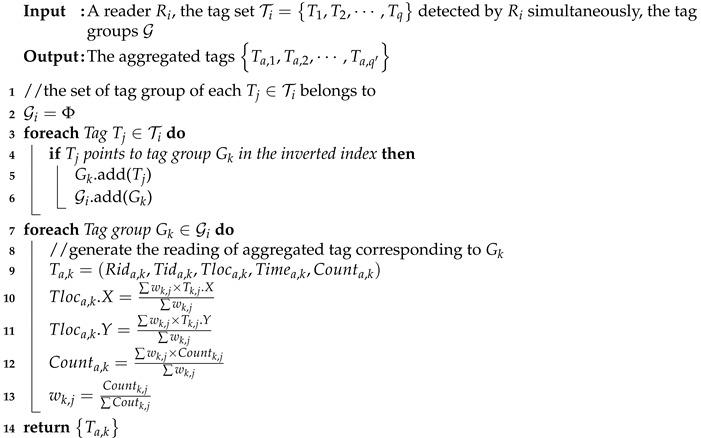


**Example** **4.**
*(Example 1 Continued) We map the distribution of ten tags detected by $R_{3}$ simultaneously at time $t_{1}$ into a 2D coordinate system, as shown in Figure 4a. Additionally, the readings of tags by $R_{3}$ is shown in Table 1. The tag groups generated based on Algorithm 2 with $\delta_{d}=35$ cm and $\delta_{s}=3$ is shown in Figure 4b. Note that the candidate tag group $G_{3}$ is removed since its size is smaller than the threshold $\delta_{s}$. Finally, based on Algorithm 3, the readings of corresponding aggregated tags are shown in Table 2.*


After aggregated tag generation, we can determine the PLR for each reader based on the aggregated tags. The pseudo-code is shown in Algorithm 4, where for each reader $R_{i}\in R$, we first generate the readings of aggregated tags based on the readings of tags detected by $R_{i}$ simultaneously (Line 3). Note that the tags which do not belong to any tag group are also used for PLR determination. Thus the tag list $T^{\prime }$ is generated by combining such independent tags and aggregated tags together to determine the PLR of $R_{i}$ (Line 5). Afterwards, the PLR of $R_{i}$ is determined by intersecting the RR of each tag in the tag list $T^{\prime }$, where the procedure is the same as the exact solution (Line 7-16). Suppose *n* tags are divided into $n^{\prime }$ tag groups. Then the time complexity of approximate solution becomes $O\left(mn^{\prime }\right)$ which is much smaller than the exact solution since $n^{\prime }\ll n$. Note that the computational cost of tag group generation is not included since it can be pre-computed once the tag deployment finished.
**Algorithm 4:** PLRGenApproximate(T, R, D, $a_{i}$, $b_{i}$, α)
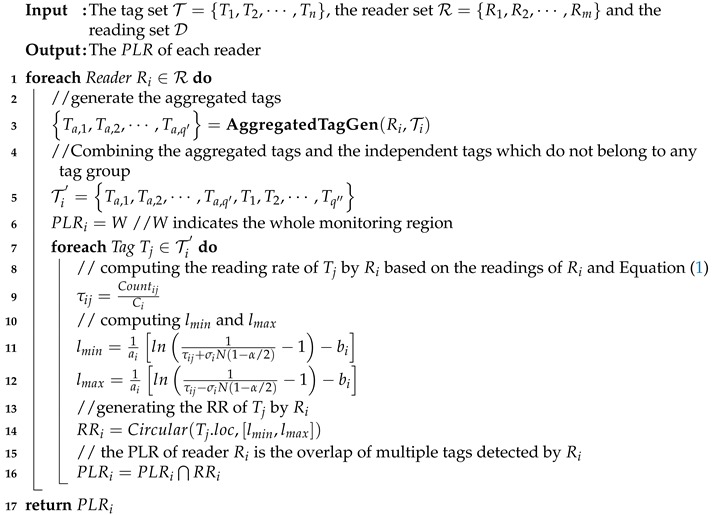


### 4.3. Location Estimation

Thus far, for each reader, we get its possible locations. In this section, we aim to estimate the precise location for each reader.

We propose a location estimation algorithm based on the grid. Given a reader $R_{i}$ which detects *q* tags, i.e., $T_{i}=\left(T_{1},T_{2},\cdots ,T_{q}\right)$, at a certain time point *t*, we first divide its PLR into grids where the grid granularity can be specified by users. Then each grid is considered as a location and denoted by its center point. Next, for each grid $g_{k}$, we compute the Root Mean Square Error (RMSE) of the reading rate of tags detected by $R_{i}$ simultaneously, by supposing that reader $R_{i}$ locates at $g_{k}$. Specifically, the RMSE of reading rate is denoted by $s_{k}=\sqrt{\frac{1}{q}\sum_{T_{j}\in T_{i}}{\left(\tau_{ij}^{{}^{\prime }}-\tau_{ij}\right)}^{2}}$, where $\tau_{ij}^{{}^{\prime }}$ and $\tau_{ij}$ are the reading rates computed based on Equations (Equation 1) and (Equation 2) respectively. Finally, we select the grid where the RMSE of reading rate is the smallest as the precise location of reader $R_{i}$, because a smaller RMSE means that the grid is closer to the real location of the reader. The pseudo-code of the algorithm is shown in Algorithm 5.
**Algorithm 5:**LocationEstimation(R, $\left\{{PLR}_{i}\right\}$)
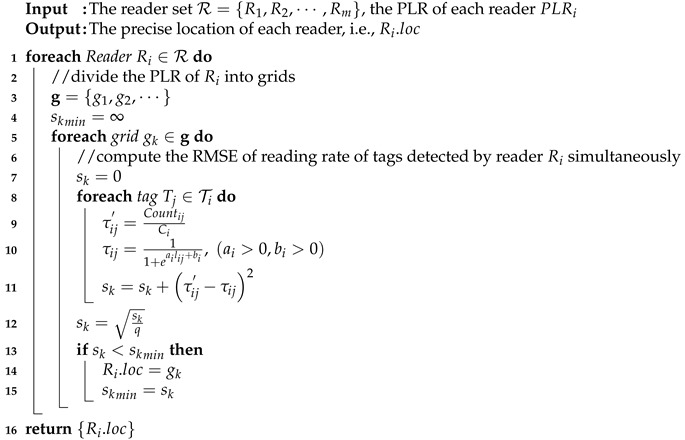


## 5. Experiments

This section reports the experimental study of the MRLIHT approach. All the experiments are performed on a PC with 3.4 GHz CPU and 16GB RAM.

### 5.1. Experimental Settings

To evaluate the performance of the proposed MRLIHT, we use the Tossim [28] which is a TinyOS [29] simulator, to simulate an application of indoor exhibition where the size of the visiting area is a 150 m × 150 m rectangular region with the grid. The Visitors (equipped with RFID readers) can move in the visiting area arbitrarily. The RFID tags are deployed on the floor of the visiting area randomly and intensively (liking the deployment shown in Figure 1 introduced in Example 1). Since in our M-RFID model, the RFID tags are fixed deployed in the visiting area, the location of each tag is known and written in its own memory. In the testbed, the type of RFID reader is supposed to be LJYZN-107 with power 915MHz and the radio range is about 10 m. The RFID tags are the labels conforming to the standard of EPC CLASSI G2. The reading rate of tags by readers is generated based on the reader detection model where the parameters are $a_{i}=3.6423$ and $b_{i}=-7.5897$. In addition, we add a random term in a range of $\left[0,0.1\right]$ on the reading rate to simulate the noise in the real application environment.

We adopt the average Euclidean distance between the real locations and estimated locations of RFID readers to measure the accuracy of various location tracking methods. Moreover, we implement two existing RSS-based indoor location tracking methods, i.e., nearest-neighbor (NN) [24,25] and multilateration (ML) [26] and a newest phase-based mobile reader localization (shorted by PHL) method proposed in [21]. It is notable that in [21], the proposed localization algorithm focuses on determining the location of a mobile RFID reader at a certain axis (e.g., x-axis) while assuming that the position on the other axis (e.g., y-axis) is known and the mobile reader moves along the certain axis. We suppose the assumption limits its application. Thus, in the experiments, we use the phase differences of the tags detected by the mobile reader to replace the RSS, and employ the multilateration to determine the final location of the moving individuals (readers). Finally, we repeat each test 100 times, where in each test we randomly plan a trajectory for each visitor, and the average results are reported to obtain reliable experimental results.

### 5.2. Selection of Parameters

Recall that the parameters $\delta_{d}$ and $\delta_{s}$ are the distance threshold and size threshold used in tag group generation introduced in Section 4.2.2. Figure 5 reports the distance deviation of MRLIHT-A by varying $\delta_{d}$ and $\delta_{s}$ under the scenarios where 50, 100, 200, 500 tags can be detected by a reader simultaneously. As shown in Figure 5a, with 50 tags can be detected by a reader simultaneously, the lowest distance deviation (3.973 cm) is achieved with $\delta_{d}=20$ cm and $\delta_{s}=3$. Since the tags are fixed deployment in the monitoring area, we can collect the test readings before the real applications and learn the best $\delta_{d}$ and $\delta_{s}$ based on the test data. In the following experiments, we set the default number of tags that can be detected by a reader simultaneously as 50 until a special illustration.

### 5.3. Comparison between our Proposed Approaches

In this section, we compare the performance of the proposed MRLIHT based on the exact and approximate PLR generation, respectively, by varying the variance σ of reading rate error ϵ and grid size.

#### 5.3.1. Varying the Variance σ of ϵ

As shown in Figure 6, with an increase of σ, the distance deviations between the estimated locations and ground truths based on both MRLIHT-E and MRLIHT-A decrease. Based on Equation (Equation 5), a large σ derives a large RR of a tag to a reader, and thereby derives a large PLR of the reader. Therefore, we can get a more precise location by checking more grids during final location estimation with a larger PLR, while we need to take more time.

On the other hand, we can observe that the accuracy of location estimation based on MRLIHT-A is slightly higher than that of MRLIHT-E, because in MRLIHT-E, we determine the PLR of a reader by using all tags it detected simultaneously without aggregating those tags into aggregated tags. Moreover, with extensive experiments, the difference between the distance deviation based on MRLIHT-E and the distance deviation based on MRLIHT-A is most 10 cm, which verifies that MRLIHT-A performs well in location tracking. However, the time cost of MRLIHT-A is much lower than that of MRLIHT-E. Since the time cost of MRLIHT-E is one order of magnitude higher than MRLIHT-A (as observed in Figure 6b), we present its time cost by using another ordinate.

#### 5.3.2. Varying the Grid Size

Figure 7 reports the accuracy of location tracking by varying the grid size. Since during location estimation, we consider a grid as a location point, the time cost decrease with the increase of grid size (as shown in Figure 7b) while the accuracy is deteriorated. As shown in Figure 7a, the distance deviations are 4.009 cm and 24.355 cm with grid sizes 10 cm and 60 cm, respectively. We set the default grid size as 60 cm to trade off the positioning accuracy and efficiency.

### 5.4. Comparison with Existing Methods

In the following experiments, we compare the proposed MRLIHT with existing methods to show that MRLIHT outperforms the state-of-the-art methods.

#### 5.4.1. Varying the Number of Tags Detected by a Reader Simultanesously

Since we estimate the location of moving individuals (readers) based on the readings of tags detected by the reader simultaneously, Figure 8 shows the performance of various methods by varying the number of tags detected by a reader simultaneously. It can be observed from Figure 8a that the lowest distance deviation is achieved by the proposed MRLIHT. The reason is that we consider the uncertainty of the reading rate of a tag by a reader and first extend the candidate locations of a reader into a PLR rather than some certain points. On the other hand, by refining each grid, the positioning accuracy is further improved. Moreover, in NN and ML, the distance between an RFID reader and tags it detected is computed based on the reading rate model whose parameters are learned based on the historical data. During the parameter learning, it inevitably derives some deviations (due to the uncertainty of the reading rate). As a result, the localization algorithms (i.e., NN and ML) without considering this uncertainty derive a large localization deviation. By comparison, in PHL, since the phases are directly collected from reference tags, the distances (between an RFID reader and tags it detected) estimated based on those phases contain relatively small deviations. Therefore, the localization accuracy of PHL is higher than those of NN and ML, but it is lower than our proposed approaches obviously.

As shown in Figure 8b, with the increase of the number of tags, it is not surprising that the time cost rises. The time costs of existing methods NN, ML and PHL (introduced in Section 5.1) are lower than those of MRLIHT-A and MRLIHT-E, but their localization accuracies are also lower. We can observe that the time costs of ML and PHL are close to each other, because both of them estimate the locations of indoor humans based on the multilateration. The main difference between ML and PHL is that the distance between an individual (reader) and a tag is computed based on the RSS and phase difference, respectively. Moreover, we observe that the time costs of the proposed approaches increase approximately linearly, which verifies that they also have good scalability. For example, the time cost of MRLIHT-A is 0.85163ms with 100 tags can be detected by a reader simultaneously, which means we can execute 6300 times of location estimation in a second. On the other hand, we can observe that the time costs of MRLIHT-A are 0.00809ms and 0.2514ms with 50 and 300 tags can be detected simultaneously, respectively. When the tag deployment is intensive, the efficiency of the proposed MRLIHT decreases. Thus in real applications, we should avoid too-intensive tag deployment with a certain accuracy guarantee.

#### 5.4.2. Varying the Number of Readers

Figure 9 reports the evaluation of the scalability in terms of accuracy and efficiency by varying the number of readers. As shown in Figure 9a, the accuracy remains stable with the increase of the count of readers. However, the accuracy of the proposed MRLIHT is higher than the existing methods, obviously. For the time costs, as shown in Figure 9b, all methods increase linearly with the increase of the number of readers. The time costs of MRLIHT-A and MRLIHT-E are 47.5284 ms and 694.6388 ms with 1000 readers, respectively. In other words, we can estimate the locations of 20,000 and 1500 readers in one second based on MRLIHT-A and MRLIHT-E, respectively. Therefore, MRLIHT-A is more suitable for the large-scale monitoring applications without much accuracy sacrifice.

## 6. Conclusions

In this paper, we have studied the indoor human tracking problem. Accordingly, we propose a novel M-RFID based localization approach for indoor human tracking, named MRLIHT. In MRLIHT, RFID tags are fixed deployed in the monitoring area intensively, and RFID readers are equipped on the moving objects (i.e., individuals) we targeting to locate. To improve the localization accuracy, we add a random disturbance term which reflects the uncertainty of the readings by an RFID reader on the reading rate. As a result, the response region of an RFID tag to the reader is extended from a circle to a circular, which enriches the candidate positions of the reader. Moreover, in MRLIHT, the individuals can move in the monitoring area arbitrarily without a pre-defined motion model, which is more applicable in the real world. In time cost, since the locations of RFID tags are fixed and can be written in their own memories, the location information of RFID tags (which are used for location tracking) can be obtained by RFID readers directly. Therefore, the extra queries for tag locations are saved, and thereby the location estimation efficiency is improved. Finally, the extensive experimental results show that the proposed MRLIHT has higher location estimation accuracy than the state-of-the-art methods. Moreover, it is able to accommodate large-scale monitoring applications with good scalability.

## Figures and Tables

**Figure 1 sensors-20-01711-f001:**
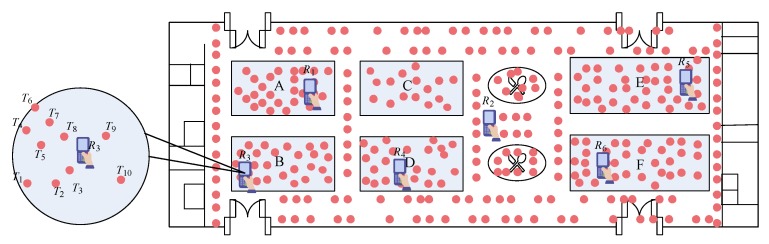
The plan of a large-scale indoor exhibition area.

**Figure 2 sensors-20-01711-f002:**
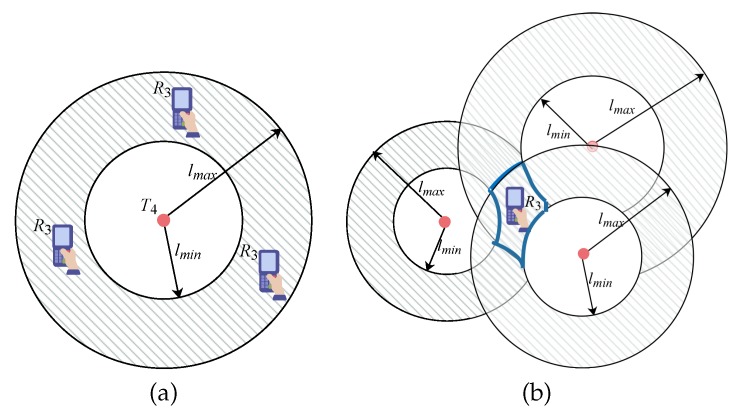
(**a**) The response region of tag $T_{4}$ to reader $R_{3}$ and (**b**) the Potential Location Region (PLR) of reader $R_{3}$.

**Figure 3 sensors-20-01711-f003:**
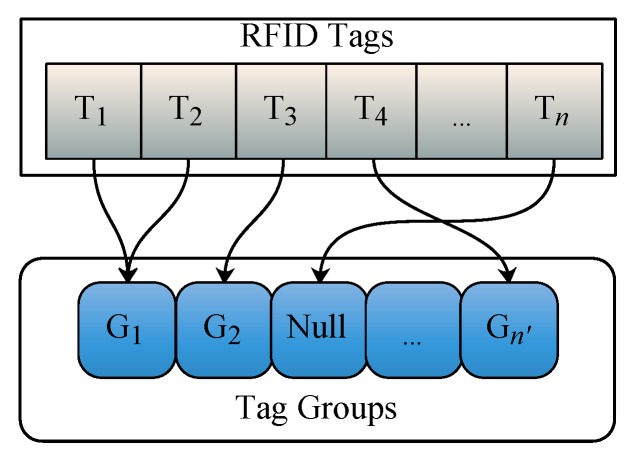
The structure of the inverted index for Radio Frequency Identification (RFID) tags.

**Figure 4 sensors-20-01711-f004:**
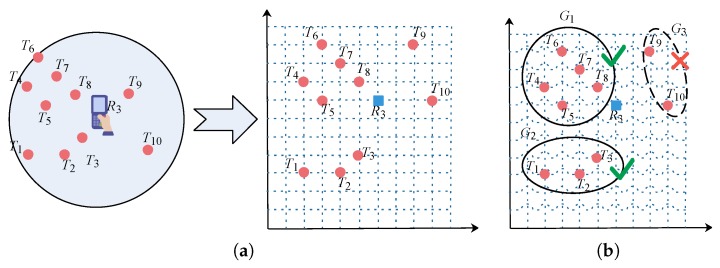
(**a**) Distribution of tags detected by $R_{3}$ and (**b**) readings of tags by reader $R_{3}$.

**Figure 5 sensors-20-01711-f005:**
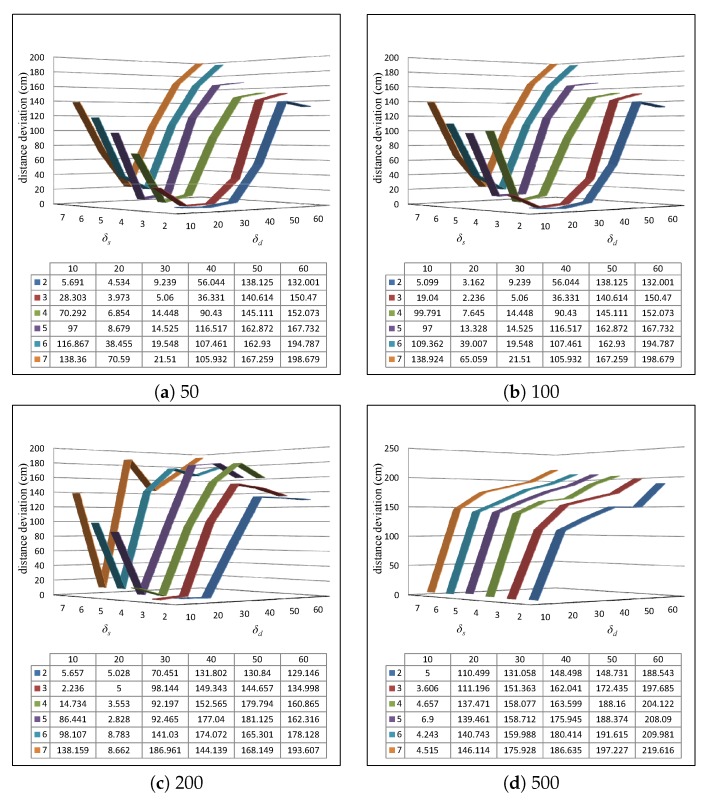
Varying $\delta_{d}$ and $\delta_{s}$ with 50, 100, 200, 500 tags can be detected by a reader simultaneously.

**Figure 6 sensors-20-01711-f006:**
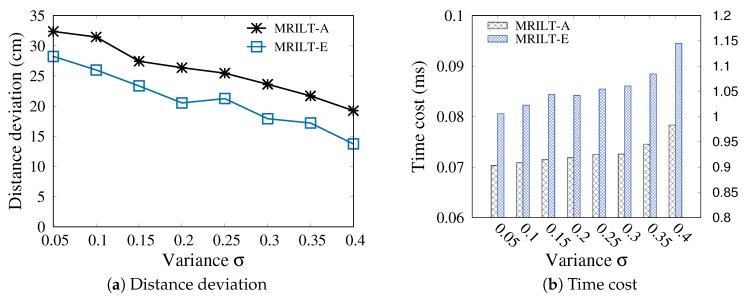
Varying σ with $\delta_{d}=20$ cm, $\delta_{s}=3$, grid_size=60 cm.

**Figure 7 sensors-20-01711-f007:**
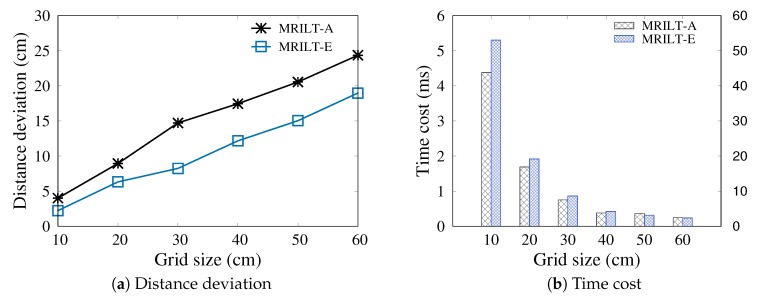
Varying grid size with $\delta_{d}=20$ cm, $\delta_{s}=3$ and σ=0.1.

**Figure 8 sensors-20-01711-f008:**
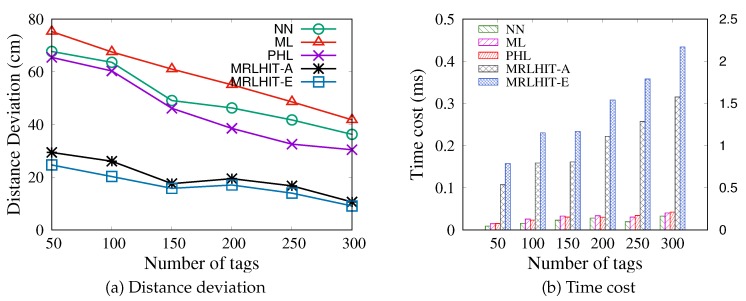
Varying the number of tags detected simultaneously with $\delta_{d}=20$ cm, $\delta_{s}=3$, grid_size = 60 cm.

**Figure 9 sensors-20-01711-f009:**
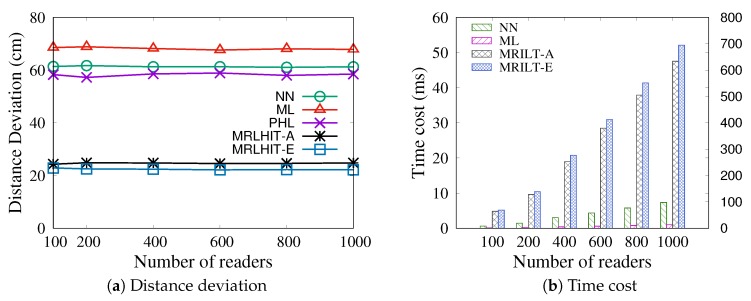
Varying the number of readers with $\delta_{d}=20$ cm, $\delta_{s}=3$, grid_size=60 cm.

**Table 1 sensors-20-01711-t001:** Readings of tags by reader $R_{3}$.

Observations	
$\left(R_{3},T_{1},\left(2,3\right),t_{1},5\right)$	$\left(R_{3},T_{6},\left(3,10\right),t_{1},6\right)$
$\left(R_{3},T_{2},\left(4,3\right),t_{1},6\right)$	$\left(R_{3},T_{7},\left(4,9\right),t_{1},10\right)$
$\left(R_{3},T_{3},\left(5,4\right),t_{1},9\right)$	$\left(R_{3},T_{8},\left(5,8\right),t_{1},20\right)$
$\left(R_{3},T_{4},\left(2,8\right),t_{1},7\right)$	$\left(R_{3},T_{9},\left(8,10\right),t_{1},8\right)$
$\left(R_{3},T_{5},\left(3,7\right),t_{1},9\right)$	$\left(R_{3},T_{1}0,\left(9,7\right),t_{1},9\right)$

**Table 2 sensors-20-01711-t002:** Readings of aggregated tags by reader $R_{3}$.

Observations
$\left(R_{3},T_{a,1},\left(3.95,3.25\right),t_{1},7\right)$
$\left(R_{3},T_{a,2},\left(3.83,8.25\right),t_{1},13\right)$

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
