# Peer review of "MRLIHT: Mobile RFID-Based Localization for Indoor Human Tracking"

_sensors, 2020, doi:10.3390/s20061711_

Round 1
Reviewer 1 Report
I think the paper is unfinished for the journal expectations.
The paper describes interesting approach for indoor location tracking. But many features are not described in enough details.
But I think it is not clearly described, for example how they repeat 100 times the test, and what was the test protocol. What device was used as the mobile terminal, a computer? Is it applicable to small mobile terminal? What was the trajectory tested? Was the results compared to some other RFID indoor localization solutions? What was the computational costs and how it could work on small terminal? As the terminal has 10m range it will have higher power consumption, was it measured? There was no problem with interference between different tags in range? How this is solved in this solution? How you physically distributed the RFID tags? So you had physical position of each RFID tag and all of them are static?
Please add comparison with other indoor location tracking solutions.
Please extend the conclusion and explain how your solution contribute to state of art solutions.
Please check references and add more details to references (pages, Venue for conference papers, etc.) , check capitalization ( wiley & sons, rfid tags
)
Reviewer 2 Report
This paper present an RFID based indoor localization system. The methods and experimental studies are competitive. However, the logic of the paper is not clear and there are overclaims. My concerns are summarized as follows:
- In the abstract “Nevertheless, existing methods are mostly developed based on the fixed RFID model where RFID readers are fixed deployed in the monitoring area and RFID tags are attached on the moving objects targeting to locate. Moreover, most existing methods estimate the locations of moving objects by using the active RFID tags (powered by on-board battery), which incurs large localization errors and high costs”.
This not true. There are many investigations with mobile readers and fixed tags. You should not ignore them. On the contrary, you need to survey the state-of-the-art and compare with them to claim your contribution.
Incorrect expression again, most RFID localization methods use passive.
- Line 26-30, ”However, the indoor location tracking based on F-RFID model has the following limitations: ”
The fixed reader and mobile reader are used for different application scenarios. For example, product items on the shelf and in distribution, mobile tag is more suitable. How many mobile readers you want to uses if you make it mobile?
Therefore, this comparison is meaningless if there is not a fixed application scenario. Just say “moving object” is not appropriate. Maybe human indoor localization can be better.
- Language problems. The following changes are suggested for consideration:
Non-contacting -> non-contact
Read rate -> reading rate
Round 2
Reviewer 1 Report
I am still not sure if the simulation was done in some simulating environment as Opnet modeler or you use real volunteers with RFID readers which follow the prepared random trajectory. How you use the data with other localization algorithms? Please make it more clear in the final online version. I assume the TinyOS was on the readers, so there was real setup of 150x150m room? You collected data and analyse with different algorithms?
Or everything is simulated, but then please describe the simulation environment more clearly in Experiment description section
OK, know I found it in the paper, please emphasize by italic may be the Tossim simulation environment and put also a reference to it, as it is Stanford university product.
http://tinyos.stanford.edu/tinyos-wiki/index.php/TOSSIM
Levis, P., Lee, N., Welsh, M. and Culler, D., 2003, November. TOSSIM: Accurate and scalable simulation of entire TinyOS applications. In Proceedings of the 1st international conference on Embedded networked sensor systems (pp. 126-137). - I think this is most cited work for that simulator
I think it should be mentioned at the beginning of the paragraph, because the Tossim environment is running on TinyOS - am I right? The paragraph is not clear from that point of view so please rewrite it for final version.
Thanks
Author Response
Response: Thank you very much for your valuable comment and I am sorry to make you confused.
I have rewritten the first paragraph in Section 5.1 to make readers are clear that the experiments are conducted on a simulated network by using the Tossim.
The rewritten paragraph is shown below:
To evaluate the performance of the proposed MRLIHT, we use the Tossim [29] which is a TinyOS [30] simulator, to simulate an application of indoor exhibition where the size of the visiting area is a 150m*150m rectangular region with grid. The Visitors (equipped with RFID readers) can move in the visiting area arbitrarily. The RFID tags are deployed on the floor of the visiting area randomly and intensively (liking the deployment shown in Figure 1 introduced in Example 1). Since in our M-RFID model, the RFID tags are fixed deployed in the visiting area, the location of each tag is known and written in its own memory. In the testbed, the type of RFID reader is supposed to be LJYZN-107 with power 915MHz and radio range is about 10m. The RFID tags are the labels conforming to the standard of EPC CLASSI G2. The reading rate of tags by readers are generated based on the reader detection model where the parameters are ai=3.6423 and bi=-7.5897. In addition, we add a random term in range of [0, 0.1] on the reading rate to simulate the noise in real application environment.
[29] Levis P, Lee N, Welsh M and Culler D. TOSSIM: Accurate and scalable simulation of entire TinyOS applications. SenSys, LA, CA, USA. 2003: 126–137.
[30] Chandanashree V C, Bhat U P, Kanade P, et al. Tinyos based WSN design for monitoring of cold storage warehouses using internet of things. ICMDCS, Vellore, India, Heidelberg, 2017:1-6.
Reviewer 2 Report
The authors have covered all my concerns, and I am happy with the current form.
Author Response
Response: Thank you very much for your support!